

# Leveraging large language models for spelling correction in Turkish

Ceren Guzel Turhan[1,2]

[1] Department of Computer Engineering, Gazi University, Ankara, Turkey
[2] Department of Cognitive Robotics, Delft University of Technology, Delft, Netherlands

## ABSTRACT

The field of natural language processing (NLP) has rapidly progressed, particularly with the rise of large language models (LLMs), which enhance our understanding of the intrinsic structures of languages in a cross-linguistic manner for complex NLP tasks. However, commonly encountered misspellings in human-written texts adversely affect language understanding for LLMs for various NLP tasks as well as misspelling applications such as auto-proofreading and chatbots. Therefore, this study focuses on the task of spelling correction in the agglutinative language Turkish, where its nature makes spell correction significantly more challenging. To address this, the research introduces a novel dataset, referred to as NoisyWikiTr, to explore encoder-only models based on bidirectional encoder representations from transformers (BERT) and existing auto-correction tools. For the first time in this study, as far as is known, encoder-only models based on BERT are presented as subword prediction models, and encoder-decoder models based on text-cleaning (Text-to-Text Transfer Transformer) architecture are fine-tuned for this task in Turkish. A comprehensive comparison of these models highlights the advantages of context-based approaches over traditional, context-free auto-correction tools. The findings also reveal that among LLMs, a language-specific sequence-to-sequence model outperforms both cross-lingual sequence-to-sequence models and encoder-only models in handling realistic misspellings.

# INTRODUCTION

Natural language processing (NLP) has become a crucial field of artificial intelligence that enables machines to not only understand and interpret languages but also generate meaningful texts. NLP studies are evolving to understand cross-linguistic structures and to solve complex tasks using large language models (LLMs) at an unprecedented speed. Indeed, the capabilities of OpenAI's GPT-4 have been assessed as meeting or surpassing the average scores of students on a graduate-level exam (*Stribling et al., 2024*). These cutting-edge models significantly impact many NLP tasks that have traditionally been the domain of humans (*Hu, 2023*; *Bubeck et al., 2023*; *Birhane et al., 2023*; *Stribling et al., 2024*). They demonstrate exceptional performance in language understanding when provided with appropriate data. However, language data sources, such as human-written texts, often contain misspellings (*Cucerzan & Brill, 2004*), which pose challenges for

Corresponding author
Ceren Guzel Turhan,
cerenguzel@gazi.edu.tr

language understanding. Both natural and synthetic spelling errors significantly affect the performance of LLMs across various NLP tasks (*Belinkov & Bisk, 2017*; *Ebrahimi et al., 2017*; *Pruthi, Dhingra & Lipton, 2019*; *Jayanthi, Pruthi & Neubig, 2020*). Therefore, spelling correction is considered an important task that focuses on the automatic detection and correction of misspellings. It is useful for many applications, such as document editing, automatic speech recognition systems, auto-proofreading, and chatbots. Search engines and text editors have provided spell check functionality for many years. In addition, open-source libraries such as GNU Aspell (*Atkinson, 2019*), JamSpell (*Ozinov, 2019*), Wikifixer NNet (*Melli et al., 2020*), and NeuSpell (*Jayanthi, Pruthi & Neubig, 2020*) are available. These libraries are generally employed with English texts and dictionaries from English Wikipedia and news websites. Unfortunately, only a few toolkits provide multilingual support. Despite Turkish being one of the top 20 most spoken languages globally, research in natural language processing for Turkish remains limited compared to more widely studied languages. As an agglutinative language, a single verb can generate over a hundred distinct forms in Turkish (*Oflazer & Saraçlar, 2018*). This means that spell checkers need to consider various forms of the same word (*e.g.*, plural, contraction) in English, whereas in languages with more complex inflections, such as Turkish, many words are formed with suffixes. Therefore, spell correction in such languages becomes more challenging (*Bhaire et al., 2015*).

Until JamSpell, spelling correction libraries focused on correcting words without considering the surrounding context. These tools, which depend mainly on dictionary-based approaches, can be viewed as context-free auto-correction tools. Considering the content has still been addressed in a limited number of studies, despite the existence of many neural models for various main NLP tasks and their downstream applications. This study aims to explore the application of LLMs for context-based spelling correction in Turkish. Additionally, this study investigates the performance of cross-lingual models for this task. Grammatical error correction is not addressed here, as it is considered a separate task in the field. Since there are currently no available benchmarks in Turkish, this study introduces a dataset by adding synthetic spelling errors to 7 million Wikipedia sentences, referred to as NoisyWikiTr. This dataset was generated by introducing character-level noise into words through insertion, substitution, deletion, and permutation of uniformly generated errors, while also simulating user-typing behaviors. Some existing pre-trained LLMs, including bidirectional encoder representations from transformers (BERT) (*Devlin, 2018*) based encoder-only models and T5 (Text-to-Text Transfer Transformers) (*Raffel et al., 2020*) based encoder-decoder models, are fine-tuned to correct misspelled sentences for the first time, as far as is known. Moreover, existing libraries that support Turkish are compared with LLM-based correctors to analyze the effect of context, alongside an n-gram-based tool that also takes content into account. Consequently, this study provides a comprehensive comparison of existing spelling correction libraries and the presented LLMs for Turkish, making a significant contribution to the field. The experimental results show that contextual models outperform context-free auto-correction tools in terms of accuracy and word-level correction. Among the large language models, the findings indicate that a language-specific sequence-to-sequence

model outperforms both cross-lingual sequence-to-sequence models and BERT-based masked language models in correcting misspellings.

## RELATED WORKS

### Spell checkers tools with multilingual support

Early studies predominantly have employed rule-based models that relied on a predefined set of rules for spelling error detection and correction, such as the SPELL system (*Pollock & Zamora, 1984*). In the following years, studies (*Norvig, 2007*; *Garbe, 2012*; *Atkinson, 2019*) have generally focused on candidate generation for correction, employing an edit distance metric along with a lookup table. The introduction of n-grams, a core concept in natural language processing, has greatly improved the ability of models to execute context-sensitive corrections (*Jurafsky & Martin, 2024*). To incorporate the context of adjacent words, JamSpell (*Ozinov, 2019*) utilizes a 3-gram language model alongside a dictionary search for effective word-level correction. Moreover, morphology analyzers have been utilized to address misspellings by generating possible word forms for unrecognized tokens through the affix rules and dictionaries. Hunspell (*Tron et al., 2005*) generates suggestions by leveraging morphological analysis rules along with dictionary-based data. By providing language-specific dictionaries for these approaches, such systems enable support for multiple languages. In the study (*Gupta, 2020*), the Symmetric Delete algorithm (*Garbe, 2012*) is applied to predict candidates for comparison with dictionaries that utilize data structures such as Tries, DAWGs (Directed Acyclic Word Graphs), and BK Trees (Burkhard-Keller Trees) (*Burkhard & Keller, 1973*). Subsequently, misspellings can be corrected using the n-gram conditional probabilities of the candidate suggestions.

### Spell checkers based on large language models for English

Relatively few studies have utilized large language models to address misspellings in English texts. *Li, Liu & Huang (2020)* involves training a vanilla transformer model (*Vaswani et al., 2017*) as a character-level encoder (*Näther, 2020*), while also integrating word-character and subword encoders to leverage character-level embeddings alongside subword representations. In Neuspell (*Jayanthi, Pruthi & Neubig, 2020*), a contextual spell correction tool is introduced that supports neural models such as Embeddings from Language Model (ELMO) (*Peters et al., 2018*), long short-term memory (LSTM) (*Hochreiter, 1997*), and pre-trained BERT, allowing for the consideration of surrounding word representations.

### Turkish morphological spell checkers

With the aim of presenting the first spelling checker tool for Turkish, a morphological root-driven parser is introduced (*Solak & Oflazer, 1993*), specifically developed for Turkish words. This parser has also been designed for spelling correction in online texts. It incorporates complex word formations and various phonetic harmony rules for the correction task, including the determination of roots morphophonemic checks, and morphological parsing. Zemberek (*Akın & Akın, 2007*) was developed as an open-source

NLP library for Turkic languages. It offers fundamental NLP functions, including spell checking, word suggestion, stemming, morphological parsing, word construction, conversion of ASCII-only text, and syllable extraction. The library relies on a root dictionary-based parser and aims to construct a specialized DAWG tree to represent the deformed states of root words as a dictionary. Based on these trees, the Zemberek spell checker corrects the given text if any matches are found by eliminating the need to search through all possible matches. In another study, a rule-based approach has been proposed to correct text by applying multiple normalization layers that address specific types of errors, aiming to obtain the canonical form of a noisy word (*Torunoğlu-Selamet & Eryiğit, 2014*). MUKAYESE (*Safaya et al., 2022*) has been introduced as a set of NLP benchmarks aimed at addressing the shortage of structured benchmarks for Turkish. Moreover, this study focuses context-free spell checking and correction for single-words, presenting a comparison among their dictionary-based Hunspell, another dictionary based Hunspell model, and the Zemberek, using their single-text correction dataset, TRSPELL10. In *Koksal et al. (2020)*, #Turki$hTweets is introduced as a publicly accessible small-scale dataset of tweets collected to form a natural noisy data. This study has evaluated two morphology-based approaches, along with edit distance and rule-based models, to provide a preliminary analysis. A sequence-to-sequence model leveraging LSTM architecture has been proposed for text normalization, with inputs derived from a character-based tokenizer that processes the first three consonant letters of misspelled words (*Büyük, 2020*). Utilizing LSTM and bidirectional LSTM (BiLSTM) networks, the study (*Aytan & Şankar, 2023*) has focused on the detection of misspelled words rather than correction by evaluating the effect of different tokenizers. In a recent study (*Oral et al., 2024*), a Turkish spelling corrector designed for e-commerce search engines was proposed. This approach utilizes a character-level transformer to generate representation-based candidates, enhancing the accuracy of search queries.

## SYNTHETICALLY CORRUPTED TURKISH DATASET: NOISYWIKITR

Due to the lack of a suitable dataset for the given task to train LLMs, which typically require a large *corpus* of misspelled and corrected sentence pairs, a spelling correction benchmark is first constructed by corrupting sentences to generate sentence pairs. For this purpose, a Turkish Wikipedia dump containing 1.6 million clean sentences is used. In the preprocessing step, a standard text-cleaning process is followed, which includes the removal of HTML tags, numbers, and special characters. Due to the unique characters in the Turkish alphabet, Unicode with Normalization Form KC Decomposition, followed by Canonical Composition (NFKC) normalization is applied to convert texts to lowercase, rather than using standard libraries for case conversion. First of all, to obtain noised and clean sentence pairs, 10% of the tokens in each sentence are corrupted by introducing noise, as in the standard word noising strategy used in *Sakaguchi et al. (2017)*, *Jayanthi, Pruthi & Neubig (2020)*. In addition, to mimic natural misspelling patterns, a realistic

noised version of the dataset is created through user-behavioral insertion, deletion, permutation, substitution, and replacement with common mistakes. For insertion and substitution in a user-behavioral manner, adjacent characters on the keyboard are considered instead of random characters, resulting in noise with a maximum edit distance of 1. For the replacement process, language-specific common mistakes are derived from predefined common misspellings, as seen in Wikipedia (https://tr.wikipedia.org/wiki/ Vikipedi:Turkcede_sik_yapilan_hatalar). Through the replacement of commonly mistaken words, the NoisyWikiTr dataset also contains misspelled words with an edit distance greater than 1. The NoisyWikiTr dataset (https://github.com/cgturhan/trspell) is publicly available and significantly larger than English datasets, containing more than ten times the number of sentences compared to the BEA60K dataset (*Jayanthi, Pruthi & Neubig, 2020*). The details of the dataset, including the types of noise and their respective counts for each data split, are provided in Table 1. A summary of the noising operations, along with examples, is presented in Table 2. For future studies, separation noise has also been added to the training dataset. Due to the word-level prediction nature of context-free tools, the correction of separation errors is not addressed in this study.

The insertion operation, $I(w)$, introduces an alphabetic character ($a \in A$) into a word at a randomly selected position, $i$, that is $\{w[0:i] + a + w[i:] \mid 0 \leq i \leq |w|$. This results in a new word. Only letters that are adjacent on a Turkish QWERTY keyboard are inserted. For example, inserting an adjacent key into "kelime" could yield words such as "kkelime", "klelime", "kelkime", "kelişme", "kelimne", and "kelimes" depending on where the insertion occurs. Moreover, vowel insertion (insertion of {a, e, i, o, u, ü, ö}) is considered due to the common user tendency to repeat vowels, thereby simulating user-generated noise. For the same word, candidate noisy words resulting from vowel insertion include "keelime", "keliime", and "kelimee".

The deletion operation, $D(w)$, involves removing one character from a word by $\{w[0:i] + w[i+1:] \mid 0 \leq i < |w|\}$ where $i$ is a random position within the word to delete the character in that position. For example, deleting a character from "kelime" might produce "elime, keime, or kelie" depending on which character is removed. Similarly, "klime, kelme, or kelim" can be obtained through vowel deletion for the same word.

The permutation operation, $P(w)$, rearranges a pair of consecutive characters ($i$ and $i+1$) within a word as $\{w[0:i] + w[i+1] + w[i] + w[i+2:] \mid 0 \leq i < |w| - 1\}$. This process involves swapping two adjacent characters. For example, permuting the characters in "kelime" could change it to "kleime" by swapping "e" and "l".

The substitution operation, denoted as $S(w)$, involves replacing a character in a word with one of its adjacent letters: $\{w[0:i] + a + w[i+1:] \mid 0 \leq i < |w|, a \in S_{adj}(w[i+1]) \subset A\}$. The substitution of the letter "e" in "kelime" could produce "krlime, kwlime, kslime, kdlime".

Moreover, language-specific common mistakes are considered to construct realistic errors for the given dataset. The word-level replacement function, $R(w) = \{\hat{w} \mid (w, \hat{w}) \in \mathcal{D}^*\}$, where $\mathcal{D}^*$ denotes a common mistakes dictionary, acts as a lookup table to replace commonly mistaken words, such as "acaip" for "acayip".

**Table 1 Description of the NoisyWikiTr dataset.**

| | NoisyWikiTr dataset | | |
| --- | --- | --- | --- |
| | Train | Valid | Test |
| #Sentences | 687 k | 85 k | 85 k |
| #Tokens | 10.4 m | 1.3 m | 1.2 m |
| #Noises | 996 k | 125 k | 130 k |
| •Deletion | 5 k | 0.6 k | 0.6 k |
| •Vowel deletion | 186 k | 23 k | 25 k |
| •Insertion | 120 k | 15 k | 16 k |
| •Vowel insertion | 70 k | 9 k | 10k |
| •Permutation | 189 k | 24 k | 26 k |
| •Substitution | 190 k | 24 k | 26 k |
| •Replacement | 216 k | 27 k | 28 k |
| •Separation | 21 k | 3 k | 0 k |
| Noise ratio (%) | 10.9 | 10.9 | 12.0 |
| Avg. edit distance | 1.53 | 1.53 | 1.42 |

# SPELLING CORRECTION METHODS

## Norvig algorithm

The Norvig algorithm (*Norvig, 2007*) relies on generating candidates based on possible edits that are one or two edits away from the word $w$. It determines the most likely spelling correction for a given word $w$ in a dictionary $\mathcal{D}$ by generating possible edits, $E_w$, through deletion, insertion, substitution and permutation, as given in Eq. (1).

$$E_w = \{I(w), D(w), P(w), S(w)\}. \tag{1}$$

Next, the algorithm generates candidate corrections, $C_w$ based on distance to these edits using Levenshtein Distance ($d_L$) (*Levenshtein, 1966*).

$$d_L(w, c) = \begin{cases} |w| & \text{if } |c| = 0, \\ |c| & \text{if } |w| = 0, \\ d_L(\text{tail}(c), \text{tail}(w)) & \text{if head}(c) = \text{head}(w), \\ 1 + \min \begin{cases} d_L(\text{tail}(c), w), \\ d_L(c, \text{tail}(w)), \\ d_L[(\text{tail}(c), \text{tail}(w)) \end{cases} & \text{otherwise} \end{cases} \tag{2}$$

$$C_w = \{c \mid \min(d_L(w, c)), c \in E_w\} \tag{3}$$

where the tail of a word $w$, denoted as tail($w$), is defined as the substring consisting of all characters except the first one. The head of a word $w$, head($w$), is the first character of $w$.

To rank these candidates, the algorithm employs a probabilistic model. The goal is to select the candidate correction $c$ that maximizes the probability of $c$ being the intended correction, given the original word $w$. This is achieved by applying Bayes' Theorem, which simplifies to maximizing the product of two probabilities: the language model probability, $P(c)$, representing the frequency with which $c$ appears in text, and the error model probability, $P(w \mid c)$, which quantifies the likelihood that $w$ is an error for $c$. As given in

**Table 2 Summary of noise operations.**

| w | Deletion D(w) | Vowel deletion | Insertion I(w) | Vowel insertion | Permutation P(w) | Substitution S(w) | Replacement R(w) |
|---|---|---|---|---|---|---|---|
| acayip | aayip | acyip | accayip | acaayip | aacyip | acatip | acaip |
| bugün | buün | bgün | bugünn | bugün | bguün | buhün | bugun |
| düzeltme | düzelme | düzltme | düzelştme | düzeltmee | düzetlme | düzektme | düzelme |

Eq. (4), an arg max function is employed to choose the candidate with the highest combined probability, considering a set of possible corrections and using these models to identify the most contextually appropriate correction $\hat{w}$ for the input word $w$.

$$\hat{w} = \arg\max_{c \in C_w} P(c \mid w) = \arg\max_{c \in C_w} P(c)P(w \mid c). \tag{4}$$

## Aspell

Aspell is a spell checker and automatic correction tool that is specifically designed for use with the GNU operating system. It generates suggestions for misspelled words by leveraging both edit distance metrics and contextual analysis, with frequency-based ranking employed to ensure the accuracy of corrections. The Aspell scoring function, $S(c)$, is based on the weighted average of edit distances between the misspelled word and entries in the lexicon, which includes a collection of words, morphological information about word forms, and additional data such as frequency and part-of-speech tags. Equation (5) computes the weighted average edit distance, $d_{wL}$, using weights and the corresponding edit distances.

$$d_{wL}(w, c) = \frac{1}{n} \sum_{i=1}^{n} W_i \cdot d_L(w, c_i) \tag{5}$$

where $W_i$ represents the weight of $i$th edit operation and $c_i$ denotes the candidate resulting from the $i$th edit. The final suggestions come from the arg min of Aspell scoring, $\arg\min_{c_i \in C} S(c_i)$.

## Hunspell

Hunspell (*Tron et al., 2005*) is an autocorrect spell checker and morphological analyzer widely utilized in numerous applications and platforms, including Mac OS X, Safari, OpenOffice, Firefox, Chrome, and Adobe products. In addition, it is integrated into various tools such as the XML Copy Editor and LyX. Hunspell supports a wide range of languages with complex morphological features. This spell checker is based on a two-part dictionary system, consisting of a dictionary $\mathcal{D}$ and an affix file. Hunspell's strength lies in its use of affix compression and morphological rules to effectively generate variations. The algorithm begins with tokenization, followed by a comparison of each token against the entries in the dictionary. When a token is not found, Hunspell generates possible word forms, $w'$, applying affix rules contained in the affix file that add or remove prefixes and suffixes after finding the root. In agglutinative languages like Turkish, affix files consist of

possible suffixes to generate candidate words. For example, the root word "gel", {"iyor", "di", "ecek"} suffixes can generate candidates like "geliyor, geldi, gelecek". It also supports compound words by breaking them down into valid subwords on the basis of its dictionaries and affix rules. Then, it performs a morphological analysis to match these generated forms with valid entries in the dictionary.

## Symmetric delete spelling correction algorithm (SymSpell)

SymSpell (*Garbe, 2018*) is an exceptionally fast spell checker library that offers performance up to 1 million times faster than traditional methods through its Symmetric Delete Spelling Correction Algorithm. It improves the Norvig algorithm by generating all potential candidates $c$ exclusively through delete operations, $c \in \{I(w)\}$. Instead of directly addressing transpositions, replacements, and insertions within the input phrase, SymSpell transforms these operations into deletions of dictionary terms. This method has proven to be significantly more efficient, as replacements and insertions are both computationally expensive and language-dependent. The candidate word set $C_w$ and the candidate dictionary set $C_d$ can be derived from the candidates $D(w)$ and $\cup_{d \in \mathcal{D}} D(d)$ that have the minimum edit distance $\min(d_{\mathrm{DL}}(w, c))$, obtained by deletion.

This technique optimizes the process of generating edit candidates and searching the dictionary for a given Damerau-Levenshtein distance ($d_{\mathrm{DL}}$) (*Damerau, 1964*) which can be computed by dynamic programming recurrence for $0 \le i \le |w|, 0 \le j \le |c|$, $d_{\mathrm{DL}}(w[i], c[0]) = i$ and $d_{\mathrm{DL}}(w[0], c[j]) = j$:

$$d_{\mathrm{DL}}(w[i], c[j]) = \min \begin{cases} d_{\mathrm{DL}}(w[i-1], c[j-1]) + \mathbf{1}_{w=c}(w, c) \\ d_{\mathrm{DL}}(w[i], c[j-1]) + 1 \\ d_{\mathrm{DL}}(w[i-1], c[j]) + 1 \\ d_{\mathrm{DL}}(w[k-1], c[l-1]) + (i-k-1) + 1 + (j-l-1) \end{cases} \tag{6}$$

Here, the indicator function $\mathbf{1}_{w=c}(w, c)$ is 1 if the condition $w = c$ holds. The arg max of candidate intersections, $\arg\max_{d \in \mathcal{D}} |C_w \cap C_d|$, yields the predicted correction.

## JamSpell

JamSpell (*Ozinov, 2019*) employs a variant of the SymSpell algorithm integrated with a 3-gram language model to incorporate contextual information, thereby enhancing performance. JamSpell is also based on the principles of Norvig. The algorithm leverages SymSpell's efficient edit candidate generation and dictionary search methods, thereby improving the performance of SymSpell. Moreover, the Bloom filter is integrated with SymSpell to store the index in an efficient manner. Deletions from the dictionary are stored in the Bloom filter. When searching for candidates, deletions are first applied to the original word up to the desired depth, similar to the method used by SymSpell. However, unlike SymSpell, insertions are performed for each deletion, and the resulting words are subsequently checked against the original dictionary. The Bloom filter is utilized to access the index of deletions and to skip insertions for deletions that are not present.

In addition, it integrates an n-gram language model to further refine and filter word-level corrections based on contextual information. Rather than employing a naive approach that calculates edit distances against a *corpus*-based dictionary, the use of n-gram

dictionaries enables the spelling corrector to consider the frequency of sequences of $n$ words occurring together, thereby enhancing predictions for out-of-vocabulary words. This involves constructing comprehensive collections of n-gram sequences, consisting of $n$ words or characters, to analyze word co-occurrences for predicting the correct words. JamSpell employs trigrams (3-grams) which are sequences of three consecutive elements from a given text to account for the context. In a trigram model, the conditional probability of the word $w_i$ can be defined by the frequencies of $w_{i-1}$ and $w_{i-2}$:

$$P(w_i \mid w_{i-2}, w_{i-1}) = \frac{f(w_{i-2}, w_{i-1}, w_i)}{f(w_{i-2}, w_{i-1})} \tag{7}$$

where $f(w_{i-2}, w_{i-1}, w_i)$ is the frequency of the trigrams $(w_{i-2}, w_{i-1}, w_i)$ in the *corpus*. For spelling correction, the context score $S$ is computed and suggestions are ranked in decreasing order of this score, returning the top $k$ suggestions. The context score of n-gram $S_n$ is given:

$$S_n = W_n \sum_{j=1}^{n-1} P(w_i \mid w_{i+j-1}) \tag{8}$$

where $W_n$ is the weight for the $n$-gram score. For this case, trigram context score ($S_3$) is computed as a context score. If some n-grams $w_{i-n}, \ldots, w_{i-1}, w_i$ were not found in the text, the frequency of occurrence of the n-gram $f(w_{i-n}, \ldots, w_{i-1}, w_i)$ will be zero and the n-gram will immediately receive a zero probability. To address this issue in JamSpell the probability of each token in a trigram, $P(w_i \mid w_{i-1}, w_{i-2})$ is considered as the product of token probabilities of all orders, $P(w_i \mid w_{i-2}, w_{i-1})P(w_i \mid w_{i-1})P(w_i)$ for smoothing the trigram probability. It also addresses the issue of high memory consumption by employing a perfect hash (*Guthrie, Hepple & Liu, 2010*) based on the Hash, Displace, and Compress (CHD) algorithm (*Belazzougui, Botelho & Dietzfelbinger, 2009*) to store information about n-grams. All of these allow JamSpell to balance computational efficiency with the accuracy of contextually appropriate spelling suggestions.

## Zemberek

Zemberek (*Akın & Akın, 2007*) is an open-source natural language processing library specifically developed for Turkish. It offers tools for various tasks, including tokenization, stemming, named entity recognition (NER), language identification, and grammatical analysis. It is designed to accommodate the suffix-based nature of Turkish. It also provides normalization as a basic spell-checking feature. Since Zemberek relies on a root word dictionary, it requires a root-finding approach. A special DAWG tree is constructed, where the roots of the tree correspond to the root words, which are used as the dictionary. As a root selector, three root selectors are employed in Zemberek. For a given word, the first selector is based on a standard, strict root selection approach, functioning as a dictionary-based top-down parser. The second, an error-tolerant parser, applies the Damerau-Levenshtein (a default similarity algorithm) or the Jaro-Winkler string similarity algorithm to select candidate roots. The final, ASCII-tolerant parser generates the root by

considering letters that are not included in the ASCII character set. To provide the best suggestions, the results are ranked based on the frequency of root word usage in Zemberek.

# CONTEXTUAL SPELL CHECK WITH LARGE LANGUAGE MODELS

## Background

### Multi-head attention

The attention mechanism models the relevance between input and output representations. It can be described as a mapping function from a query $Q$ and key-value pairs $(K, V)$ to a representation where each key has dimension $d_k$ and each value has dimension $d_v$:

$$\text{Attention}(Q, K, V) = \text{softmax}\left(\frac{QK^T}{\sqrt{d_k}}\right)V. \tag{9}$$

The multi-head attention layer performs a linear transformation on the concatenated outputs of multiple attention heads, each of which represents a distinct subspace of the representation:

$$\text{Multi}-\text{Head}(Q, K, V) = \text{Concat}(\text{head}_1, \text{head}_2, \dots, \text{head}_h)W^O \tag{10}$$

where the matrix $W^O$ is a learned linear transformation applied to the concatenated output of all attention heads. Each attention $\text{head}_i$ is computed as below:

$$\text{head}_i = \text{Attention}(QW_i^Q, KW_i^K, VW_i^V) \tag{11}$$

Here, the matrices $W_i^Q$, $W_i^K$, and $W_i^V$ are the learned projection matrices for the $i$-th attention head. This approach enables the model to capture information from various representational subspaces and positions.

### Layer normalization

Layer normalization in BERT normalizes the input vector $x_i$ by computing the mean $\mu$ and variance $\sigma^2$ of the vector's elements:

$$\text{LayerNorm}(x_i) = \gamma \cdot \frac{x_i - \mu}{\sqrt{\sigma^2 + \varepsilon}} + \beta \tag{12}$$

where $\varepsilon$ is a small constant for numerical stability, and $\gamma$ and $\beta$ are learned scaling and shifting parameters. This process ensures consistent distribution of inputs, aiding in stabilizing and accelerating training.

### Feed-forward neural network

In each transformer block, a fully connected feed-forward neural network (FFNN) follows the multi-head attention layer. As a sublayer, it is defined as:

$$\text{FFNN}(x) = W_2\text{max}(0, W_1x + b_1) + b_2. \tag{13}$$

This network includes an intermediate rectified linear unit (ReLU) activation function and consists of two linear transformations applied to the input $x$. The parameters $W_1$, $W_2$, $b_1$, and $b_2$ are the weights and biases of the feed-forward neural network.

### *Bidirectional encoder representations from transformers (BERT)*

BERT (*Devlin, 2018*) is an encoder-only model built with stacked transformer blocks. Each transformer block includes two main sublayers: a multi-head attention mechanism and fully connected feed-forward layers. Each transformer layer starts with a multi-head attention mechanism which allows the model to simultaneously focus on different parts of the input sequence. This is followed by a residual connection (*He et al., 2016*) and layer normalization where the residual connection adds the initial input of the attention mechanism to its output. The next step involves processing this output through a feed-forward network, which includes two linear transformations with a ReLU activation function in between. Then, another residual connection and layer normalization are applied after the feed-forward network. The final output is then passed to the next transformer layer to further enhance the representations at each level, which are subsequently utilized for various downstream NLP tasks.

BERT is designed to serve as both a pre-training model for unsupervised tasks and a fine-tuning model for various downstream applications. During the pre-training phase, BERT was trained on unlabeled data using two unsupervised learning objectives: Masked language modeling (MLM) and next sentence prediction. There are two primary versions of the BERT model: $BERT_{base}$ and $BERT_{large}$. The $BERT_{base}$ model consists of 12 transformer layers, each with a hidden size of 512. The block representation of the $BERT_{base}$ model, shown in Fig. 1, illustrates the components of each transformation layer. The $BERT_{large}$ model, however, doubles the number of layers and hidden size of each layer, resulting in a total of 340 million parameters.

### Masked language modeling based model

In the MLM task, 15% of the tokens were randomly masked allowing the model to predict the masked words, which were replaced with a [MASK] token. For MLM, the objective is to predict these missing words based on the context provided by the surrounding words. Pre-training on masked word prediction enables models to gain a deep understanding of language context. This understanding can then be leveraged for other NLP tasks through fine-tuning.

In this study, misspelling correction is treated as a masked prediction, utilizing a pre-trained BERT-based MLM which is demonstrated in Fig. 2. As seen in the given figure, [MASK] tokens should be generated before feeding the inputs into the BERTurk encoder model. To generate [MASK] tokens for out-of-vocabulary (OOV) words, the study utilizes the Turkish transformer model from spaCy (https://spacy.io/, https://huggingface.co/turkish-nlp-suite) (*Altinok, 2023*). This work adapts the BERTurk model (*Schweter, 2020*) into the ContextualSpellCheck (https://github.com/R1j1t/contextualSpellCheck) (*Goel, 2021*) as a MLM. The input sequence $S = (w_1, w_2, \ldots, w_n)$ is masked in accordance with the existence of the word $w_i$ in the dictionary $\mathcal{D}$ using spaCy:

$$S[i] = [MASK], \forall w_i \in S \text{ such that } w_i \notin \mathcal{D}, \quad 0 \le i \le |S| \tag{14}$$

$$S = (w_1, w_2, \ldots, [MASK], \ldots, w_n) \tag{15}$$

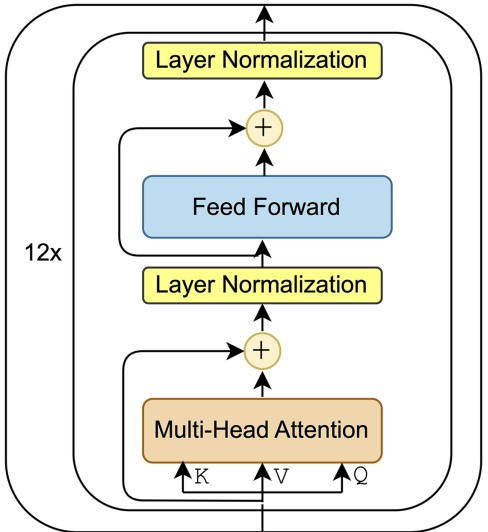

**Figure 1  Block representation of BERT$_{base}$ model.**

By adapting the ContextualSpellCheck pipeline, input sequences with [MASK] tokens are provided to a MLM, referred to as TransformerMLMEncoder for the remainder of the article, yielding embeddings for the correction task. Subsequently, a FFNN follows the TransformerMLMEncoder. This network acts as a linear layer to generate the logits for the sequence tokens. The softmax function transforms the logits into probabilities. Candidates $C_w$ are the top $k$ predictions sorted by their highest probabilities for correction. Accordingly, ContextualSpellCheck is revised to suggest a word if there is a candidate $C_i$ with an edit distance of 1. In the other case, the model outputs the given input token to prevent false positive predictions. The model based on the masked language encoder is presented below:

$$h = \text{TransformerMLMEncoder}(S) \tag{16}$$
$$\text{FFNN}(h) = W_s h + b_s \tag{17}$$
$$P(S) = \text{softmax}(\text{FFNN}(h)) \tag{18}$$
$$C_{w_i} = \text{Sort}(P(S_i), \text{'descending'})[:k] \tag{19}$$
$$\hat{w}_i = \begin{cases} c \mid \min(d_{DL}(w_i, c)) \ \forall c \in C_{w_i}, & \text{if } \min(d_{DL}(w_i, c)) \leq 1 \\ w_i, & \text{else} \end{cases} \tag{20}$$

where $W_s$ and $b_s$ are the weights and bias for the linear layer. $P(S_i)$ denotes the probability of the $i$th logit.

### Subword encoder based model

Encoder-only LLMs are also utilized with a FCNN layer to correct sentences by regenerating whole subwords. The comparison of this model with MLM based correction model is visualized in Fig. 3. Rather than predicting [MASK] tokens, the encoder predicts each subword as a token in the given sequence. The encoder of this model will be referred to as TransformerEncoder. TransformerEncoder encodes a sequence of subword tokens $S$

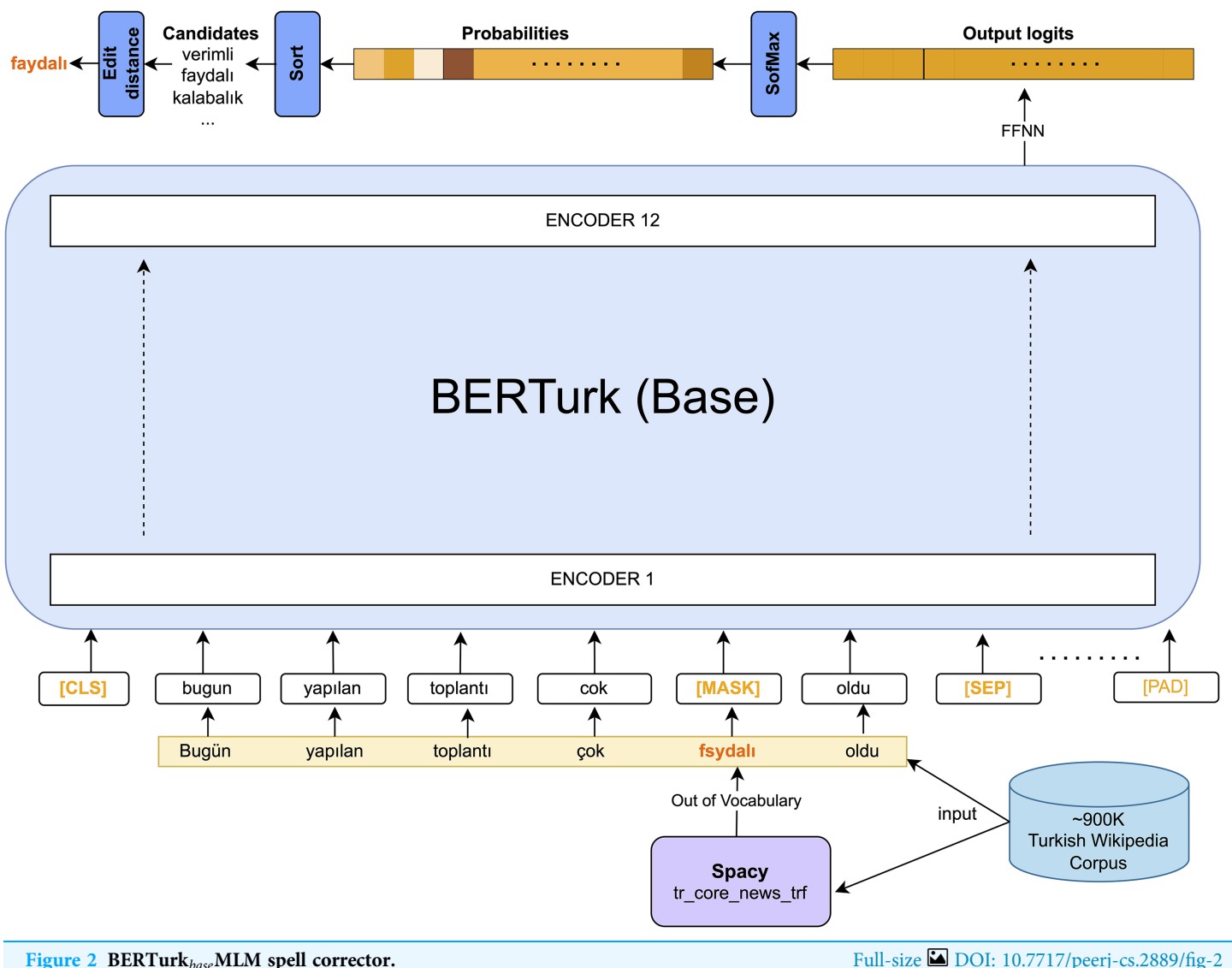

**Figure 2** BERTurk$_{base}$MLM spell corrector.

to $h_e$ embeddings. Dropout (*Srivastava et al., 2014*) is applied within the TransformerEncoder for regularization. Thereafter, subword embeddings, $h_{sub}$, are obtained by concatenating the embeddings of the subwords $w^i_{sub_k}$ in the given sequence $S$ if a token is a subword of $w_i$, as follows:

$$h_e = \text{TransformerEncoder}(S) \tag{21}$$
$$h = \text{concat}(h_{e_0}, \ldots, h_{e_k}) \tag{22}$$

where $h_{e_k}$ denotes the embeddings of $w_{e_k}$.

Subsequently, the FFNN, a single linear layer, predicts the logits of the merged token embedding. These logits are converted to probabilities, $P(S)$, by applying the softmax function to generate predictions for the words $w_i$ in the sequence $S$. The corrected sequence $\hat{S}$ is derived from the argmax of $P(S)$.

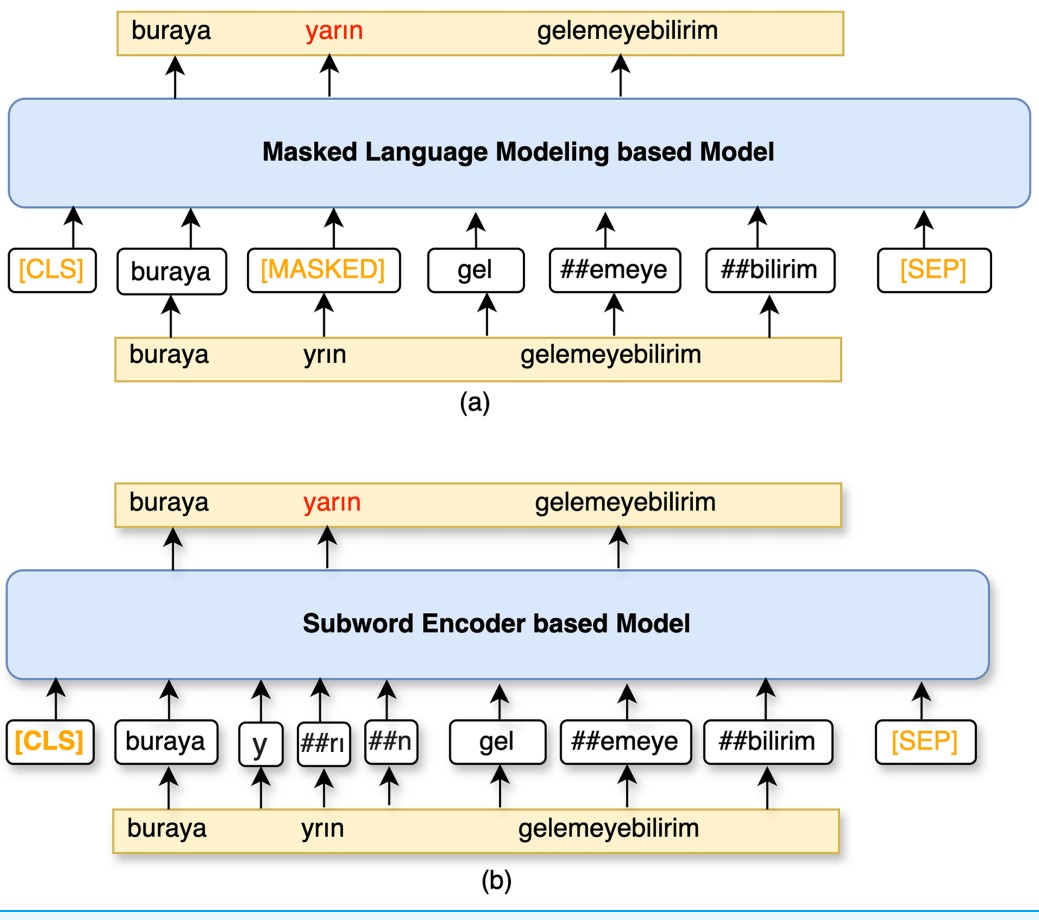

**Figure 3** A schematic illustration of (A) MLM based model and (B) subword encoder based model.

$$\text{FCNN}(h) = W_s h + b_s \tag{23}$$
$$P(S) = \text{softmax}(\text{FCNN}(h)) \tag{24}$$
$$\hat{S} = \text{argmax}(P(S)) \tag{25}$$

## Encoder-decoder LLM based model

The sentence spelling correction task will also be treated as a sequence-to-sequence generation task using pairs of noised and correct sentence pairs. Consequently, encoder-decoder LLMs pre-trained for downstream NLP tasks, such as machine translation, question answering, and next sentence prediction, are also included in this study.

The encoder-decoder LLMs have architectures derived from the framework proposed by *Vaswani et al. (2017)*. The encoder consists of multiple blocks, each containing two components: a self-attention layer and a feed-forward neural network. Self-attention layers are bidirectional, similar to those in the BERT model. Therefore, the encoder model can be referred to as a bidirectional encoder. The decoder-only model can function as a LLM, specifically a model trained for next sentence prediction like OpenAI GPT (*Radford, 2018*).

GPT relies on a left-to-right architecture, enabling it to attend only to the tokens from previous steps within the self-attention layer. In an autoregressive decoder like GPT, a token is sampled from the predicted distribution of the decoder, and then this token is fed back into the decoder to generate the prediction for the subsequent output at each time step. An encoder-decoder model combines left-to-right autoregressive language model and bidirectional encoder with MLM objective as given in Fig. 4. The decoder in such a sequence-to-sequence model has a similar structure to that of the encoder. In the default small-sized encoder-decoder LLM, both the encoder and decoder consist of six blocks. Furthermore, they have similar sizes and configurations as well.

In a sequence-to-sequence model designed for sentence spelling correction, the encoder, referred to as BidirectionalEncoder in Eq. (26), takes a sequence $S$ that may contain erroneous words as input while the decoder, namely AutoregressiveDecoder, produces a corresponding denoised sequence in an autoregressive manner. Initially, the bidirectional encoder maps an input sequence of tokens to the to the sequence of embeddings $h$:

$$h = \text{BidirectionalEncoder}(S) \tag{26}$$

$$\widehat{S}_t = \text{AutoregressiveDecoder}(S_{<t}, h) \tag{27}$$

where $S_{<t}$ means a subsequence of $S$ at time step $t$ and $\widehat{S}_t$ is the predicted subsequence. In AutoregressiveDecoder, $\widehat{S}_t$ is generated by applying the softmax function to the logits at each time step $t$.

## EXPERIMENTS

This section provides a comprehensive comparison of dictionary and edit distance-based models, n-gram-based models, and LLMs for sentence correction in the Turkish language. The performances of recent spelling correction tools and large language models, trained for the first time in this study for the Turkish spelling correction task, are analyzed and compared to assess their effectiveness.

Firstly, open-source tools supporting the Turkish language, based on dictionaries, edit distance, and n-grams, are evaluated to produce preliminary results. Notably, the GNU Aspell, Hunspell, SymSpell, Zemberek, Norvig, and JamSpell libraries are utilized as recent tools. For GNU Aspell, the aspell-python wrapper (https://github.com/WojciechMula/aspell-python) version 1.15 is employed with the Turkish dictionary package (http://aspell.net/) version 0.50-0. For Hunspell, the pyhunspell library version 0.55, which serves as python bindings for Hunspell, is used with the language-specific dictionary and affix files. For Hunspell, two existing Turkish dictionary and affix files are evaluated for comparison. Hunspell$_{\text{TDD}}$ (https://github.com/tdd-ai/hunspell-tr) (*Safaya et al., 2022*) refers to the Hunspell dictionary developed by the Turkish Data Depository (TDD) group, while Hunspell$_{\text{VDemir}}$ (https://github.com/vdemir/hunspell-tr) is another Turkish dictionary. The symspellpy library (https://github.com/mammothb/symspellpy) version 6.7.8 is used as SymSpell tool with a custom dictionary. The Turkish dictionary for SymSpell has been created from the NoisyWikiTr training set. Zemberek, an NLP tool for Turkish, is also employed in the experiments as a baseline using the zemberek-python (https://github.com/loodos/zemberek-python) version 0.2.3, a python port of the

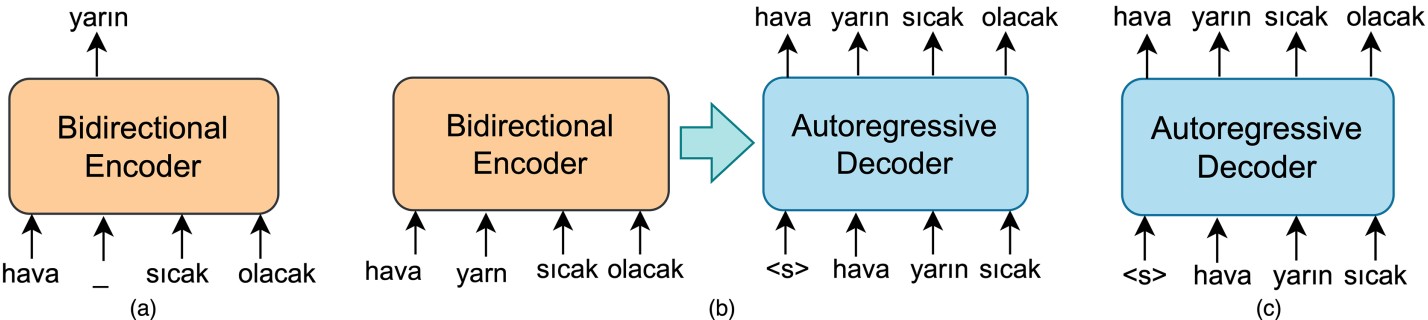

**Figure 4** A schematic illustration of (A) bidirectional encoder MLM based model and (B) autoregressive decoder model (C) encoder-decoder LLM based model.

Zemberek library, zemberek-nlp (https://github.com/ahmetaa/zemberek-nlp). The autocorrect library (https://github.com/filyp/autocorrect) based on Norvig's algorithm is another reference that supports multiple languages, including Turkish. The autocorrect library will be referred to as Autocorrect$_{Norvig}$ for the rest of the article. The JamSpell library (https://github.com/bakwc/JamSpell) serves as a multilingual spell-checking tool considering the surrounding words. In JamSpell, a language model has been trained using the Turkish alphabet and texts from the training set to generate n-grams for contextual knowledge. In the experiments, JamSpell version 0.0.12 was employed as a contextual corrector tool.

Moreover, the performance of LLMs as spell corrector is also discussed in this study. Among LLMs, the ContextualSpellCheck demonstrates that pre-trained encoder-only models can be directly utilized to correct out-of-vocabulary words in a masked prediction manner. In the original implementation, the BERT model with MLM generates candidates for unknown and OOV words in a sentence with the spaCy English pipeline leveraging contextual information. For NLP tasks in Turkish, the BERTurk model was introduced by fine-tuning the BERT$_{base}$ model for various downstream tasks, including part-of-speech tagging, named entity recognition, question answering, and MLM. This model was trained on a Turkish Wikipedia dump, a subset of the Turkish OSCAR *corpus*, various OPUS corpora, and a 35 GB *corpus*. Inspired by ContextualSpellCheck library, BERTurk and a spaCy Turkish model are used for the first time with Turkish sentences. This BERTurk$_{base}$ model is utilized using ContextualSpellCheck library as an encoder-only model in this study referring BERTurk$_{base}$MLM. In the experiments, candidate tokens predicted by BERTurk$_{base}$MLM are restricted to a minimum length of three characters during prediction due to prevent the inclusion of common and short words like "ve" (which means "and" in English).

Similar to NeuSpell (https://github.com/neuspell/neuspell) (*Jayanthi, Pruthi & Neubig, 2020*) which uses BERT for spelling correction in English, the BERTurk model is adapted for correcting Turkish texts. By modifying Neuspell for other encoder-only LLMs as subword models, such as BERTurk$_{base}$ and mBERT$_{base}$, which stands for the multilingual version of BERT$_{base}$, it becomes possible to fine-tune other encoder-only models with

byte-level byte-pair encoding (BPE) tokenizer (*Sennrich, Haddow & Birch, 2015*) for Turkish. Thus, XLM-ROBERTa (*Conneau, 2019*), a multilingual version of ROBERTa (*Liu, 2019*) designed to enhance cross-lingual language understanding, is fine-tuned for the sentence correction objective, referring to the XLM-ROBERTa$_{base}$ model. In the original Neuspell implementation, long sentences are not considered valid due to the maximum token limits in these models. To evaluate such texts in the NoisyWikiTr dataset, long sentences are split into multiple shorter sentences, and then the results are concatenated. All mentioned encoder-only LLMs are adopted from the Hugging Face library (https://huggingface.co/). The encoder-only LLMs employed in this study are BERTurk$_{base}$ (https://huggingface.co/dbmdz/bert-base-turkish-128k-uncased), mBERT$_{base}$ (https://huggingface.co/google-bert/bert-base-multilingual-uncased), and XLM-ROBERTa$_{base}$ (https://huggingface.co/FacebookAI/xlm-roberta-base).

This article also examines the performance of encoder-decoder LLMs for the given downstream task. Specifically, the T5-efficient model (*Tay et al., 2021*), mT5 (*Xue et al., 2020*), and FLAN-T5 (*Chung et al., 2024*) pre-trained models have been adapted from Hugging Face as T5-based sequence-to-sequence models. Details of the LLMs employed in this ablation study are outlined in Table 3. Meanwhile, Table 4 demonstrates the hyperparameters for these models. The maximum sequence length is set to 256 for 128 k tokens, as the average token length in the training data is 32.52, eliminating the need for additional padding and enabling faster training of T5-based encoder-decoder models. The experiments were conducted using various NVIDIA GPUs, specifically a single A100, a single L4, and two T4 GPUs, to fine-tune all large language models simultaneously. In the experiments, encoder-decoder models are trained using the ADAM optimizer (*Kingma, 2014*) with a default learning rate of 5e−5, while encoder-only models are trained using BertAdam, which is AdamW (*Loshchilov & Hutter, 2017*), with a fixed weight decay of 0.01, a warmup rate of 0.1, and a linear decay of the learning rate (5e−5). The T5-efficient-small model, fine-tuned for Turkish and referred to as T5-efficient-TR$_{small}$ (https://huggingface.co/Turkish-NLP/t5-efficient-small-turkish), is also specifically trained for the spelling correction task, using the specified hyperparameter ranges. The small-sized mT5$_{small}$ model from Hugging Face (https://huggingface.co/google/mt5-small) is employed as a cross-lingual sequence-to-sequence model for the given task. The FLAN-T5 model with a small-sized architecture, FLAN-T5$_{small}$ (https://huggingface.co/google/flan-t5-small), is trained on over 1,000 tasks and supports a wide range of languages; it is also employed in this study to address the spelling correction task. During testing, a beam size of 5 is utilized for sequence decoding across all encoder-decoder LLMs, allowing these models to explore multiple candidate sequences by maintaining the top 5 most likely options at each decoding step.

## Evaluation metrics

The comparisons are presented in terms of word correction, word-level accuracy, and Levenshtein similarity, considering the input sentence $S$, the predicted sequence $\hat{S}$, and the

**Table 3 Overview of LLMs for sentence spelling correction.**

| Model | #Params | Model architecture | Pre-training task |
|---|---|---|---|
| BERTurk$_{base}$ | 283 M | Encoder-only | Masked language modeling |
| mBERT$_{base}$ | 266 M | Encoder-only | Multilingual masked language modeling |
| XLM-ROBERTa$_{base}$ | 376 M | Encoder-only | Multilingual masked language modeling |
| FLAN-T5$_{small}$ | 80 M | Encoder-decoder | Span corruption |
| T5-efficient-TR$_{small}$ | 143 M | Encoder-decoder | Span-based masked language modeling |
| mT5$_{small}$ | 300 M | Encoder-decoder | Multilingual masked language modeling |

target sequence $T$. For the noisy word $\tilde{w}_k$ in a sentence $S$ and the corresponding denoised word and target pair $(\hat{w}_i, t_i)$, the word correction is defined as:

$$\text{Word Correction} = \frac{\sum_k \mathbf{1}(\tilde{w}_k)}{|\widetilde{W}|}, \forall \tilde{w}_k \in \widetilde{W} \tag{28}$$

where $\widetilde{W}$ is a set of noisy words in the sentence $S$. The correction indicator $\mathbf{1}(\widetilde{w_k})$: where $\widehat{w}_i \in \hat{S}$ and $t_i \in T$.

Word-level correction focuses on whether a noisy word is corrected, which may lead to a lack of consideration for the overall sentence. Therefore, sentence accuracy is also considered to evaluate the entire sentence by assessing a model's ability to retain error-free words without making any modifications. For $\forall w_i \in S$, accuracy can be computed by counting all correct words, regardless of whether they are erroneous, using the indicator function:

$$\mathscr{C}(w_i) = \begin{cases} 1, & \text{if } \mathbf{1}(w_i) = 1 \\ 0, & \text{else} \end{cases} \tag{29}$$

$$\text{Accuracy} = \frac{\sum_i \mathscr{C}(w_i)}{|S|} \tag{30}$$

For considering the similarity between predicted sentences and targets, even if without correcting full word-level misspellings, Levenshtein similarity, which depends on the Levenshtein distance ($d_L$), is also provided:

$$\text{Levenshtein similarity} = 1 - \frac{d_L(S, T)}{T} \tag{31}$$

## Experimental results

In this section, a comparative evaluation of various spelling correction tools and LLMs for the Turkish spelling correction task is presented. The performance of these models is assessed based on word-level accuracy and word correction metrics. The evaluation is conducted on the NoisyWikiTr dataset with two distinct noising strategies: uniformly generated noising and realistic noising, providing a comprehensive analysis of their effectiveness in handling different types of noise.

Table 5 presents the comparative evaluation of spelling correction tools and LLMs for the Turkish spelling correction task, based on word-level accuracy and word correction metrics, evaluated on both the uniformly generated noisy and realistically noisy sets.

**Table 4 Hyperparameter configuration for fine-tuning large language models.**

| Model | Batch size | Learning rate | Steps |
|---|---|---|---|
| BERTurk$_{base}$ | 16 | 5e−5 (warmup = 0.1) | 429 k |
| mBERT$_{base}$ | 16 | 5e−5 (warmup = 0.1) | 386 k |
| XLM-ROBERTa$_{base}$ | 16 | 5e−5 (warmup = 0.1) | 401 k |
| FLAN-T5$_{small}$ | 16 | 5e−5 | 429 k |
| T5-efficient-TR$_{small}$ | 16 | 5e−5 | 515 k |
| mT5$_{small}$ | 16 | 5e−5 | 429 k |

### Analyzing context-free and contextual models in correction

A comparative analysis of dictionary-based, n-gram models, and transformer architectures is conducted to evaluate their performance, taking into account both their context-free and contextual characteristics. According to Table 5, Aspell, Hunspell variations, and SymSpell exhibit relatively lower performance in terms of accuracy and word-level correction metrics. Although dictionary-based models focus solely on finding the closest terms without considering the entire sentence, as context-free models do, Zemberek and Autocorrect outperform other dictionary-based tools and exhibit performance that is relatively close to encoder-only models on the uniformly noised set. For the same set, JamSpell also outperforms the transformer models. When using the pre-trained BERTurk$_{base}$ model as an MLM, BERTurk$_{base}$MLM demonstrates lower performance compared to other encoder-only models, showing relatively low word correction rates. When considering the denoising capability of sequence-to-sequence models for the uniformly noised set, it is observed that character perturbations significantly negatively impact the performance of these models, as demonstrated by *Belinkov & Bisk (2017)*. In contrast, for the realistically noised set, sequence-to-sequence models yield promising results, as they are better equipped to handle natural language variations that more accurately reflect real-world misspellings. Overall, contextual models outperform context-free models, demonstrating a significant margin in performance. This highlights the superiority of models that incorporate contextual information over those relying solely on context-free approaches.

### Comparison of denoising performances for realistic and uniformly generated noises

An additional evaluation is conducted to examine the impact of the noising strategy on model performance. While dictionary-based tools like Zemberek and Autocorrect outperform encoder-decoder models in correcting uniformly noised texts, all transformer models show improved performance in handling realistic noise. For uniformly noised texts, the results show that the JamSpell model achieved the highest accuracy by a significant margin compared to other tools in terms of accuracy and word correction. The BERTurk$_{base}$ model ranks just below JamSpell, exhibiting comparable performance in word-level accuracy, but slightly underperforming in word correction accuracy. For uniformly noised texts, the encoder-only models outperform the encoder-decoder models,

**Table 5 Performance evaluation of models on the NoisyWikiTr dataset.**

| Model/Library | Uniformly generated noising | | Realistic noising | |
|---|---|---|---|---|
| | Accuracy | Word correction | Accuracy | Word correction |
| Aspell | 0.4856 | 0.3135 | 0.4492 | 0.2606 |
| Hunspell$_{TDD}$ | 0.6887 | 0.4187 | 0.6881 | 0.3793 |
| Hunspell$_{VDemir}$ | 0.7020 | 0.4886 | 0.7000 | 0.4336 |
| Zemberek | 0.9173 | 0.6130 | 0.8770 | 0.4740 |
| Autocorrect$_{Norvig}$ | 0.9238 | 0.6839 | 0.9138 | 0.5123 |
| JamSpell | 0.9606 | 0.8156 | 0.9500 | 0.6735 |
| FLAN-T5$_{small}$ | 0.7013 | 0.4192 | 0.7189 | 0.5483 |
| T5-eifficient-TR$_{small}$ | 0.7015 | 0.4185 | 0.9688 | 0.8129 |
| mT5$_{small}$ | 0.7083 | 0.4658 | 0.9662 | 0.7449 |
| BERTurk$_{base}$MLM | 0.8623 | 0.3401 | 0.8670 | 0.2378 |
| BERTurk$_{base}$ | 0.9525 | 0.6442 | 0.9593 | 0.6538 |
| XLM-ROBERTa$_{base}$ | 0.9286 | 0.4478 | 0.9421 | 0.5003 |
| mBERT$_{base}$ | 0.9315 | 0.5845 | 0.9638 | 0.6844 |

**Table 6 Performance of models in correcting language-filtered texts.**

| Model/Library | Langauge-filtered dataset | | |
|---|---|---|---|
| | Accuracy | Word correction | Levenshtein similarity |
| Aspell | 0.4883 | 0.2938 | 0.8459 |
| Hunspell$_{TDD}$ | 0.8455 | 0.4539 | 0.9528 |
| Hunspell$_{VDemir}$ | 0.7974 | 0.4850 | 0.9341 |
| SymSpell | 0.8951 | 0.4798 | 0.9783 |
| Zemberek | 0.9093 | 0.4936 | 0.9749 |
| Autocorrect$_{Norvig}$ | 0.9141 | 0.5135 | 0.9803 |
| JamSpell | 0.9511 | 0.6673 | 0.9868 |
| FLAN-T5$_{small}$ | 0.7033 | 0.5593 | 0.9422 |
| mT5$_{small}$ | 0.9582 | 0.7401 | 0.9918 |
| T5-efficient-TR$_{small}$ | 0.9664 | 0.8142 | 0.9925 |
| BERTurk$_{base}$MLM | 0.8902 | 0.2348 | 0.9691 |
| BERTurk$_{base}$ | 0.9596 | 0.6498 | 0.9755 |
| XLM-ROBERTa$_{base}$ | 0.9423 | 0.4912 | 0.9857 |
| mBERT$_{base}$ | 0.9641 | 0.6810 | 0.9780 |

including T5-efficient-TR$_{small}$, FLAN-T5$_{small}$, and mT5$_{small}$. For realistically noised texts, T5-efficient-TR$_{small}$ achieves the highest performance in both word-level accuracy and word correction. Contextual models exhibit an improvement of approximately 0.30 in word correction compared to context-free models. It can be concluded that LLMs demonstrate superior performance over context-free tools, particularly when handling more realistic noise.

**Table 7 Model performances according to noise type.**

| Noise type | FLAN-T5$_{small}$ | mT5$_{small}$ | T5-efficient-TR$_{small}$ | BERTurk$_{base}$ | XLM-ROBERTa$_{base}$ | mBERT$_{base}$ |
|---|---|---|---|---|---|---|
| Deletion | 0.0061 | 0.4080 | 0.5000 | 0.0399 | 0.0460 | 0.0552 |
| Vowel deletion | 0.4911 | 0.6852 | 0.7633 | 0.5962 | 0.4324 | 0.6305 |
| Insertion | 0.5441 | 0.7461 | 0.8191 | 0.5669 | 0.4274 | 0.6099 |
| Vowel insertion | 0.7151 | 0.8252 | 0.8715 | 0.6983 | 0.5269 | 0.7518 |
| Substitution | 0.3915 | 0.5472 | 0.6832 | 0.5347 | 0.3934 | 0.5727 |
| Transposition | 0.4910 | 0.6652 | 0.7436 | 0.5810 | 0.4382 | 0.6263 |
| Replacements | 0.7639 | 0.9605 | 0.9822 | 0.8728 | 0.6888 | 0.8726 |

### Evaluating cross-lingual vs language-specific LLMs for spelling correction

To compare cross-lingual and language-specific LLMs for the given task, a language-filtered subset of the realistically noised dataset is created by filtering and removing 870 non-target language texts using Langdetect (https://github.com/Mimino666/langdetect), a library ported from Google's language detection tool. Table 6 presents the results based on the filtered dataset. In addition to word-level accuracy and word correction metrics, Levenshtein similarity is also provided to evaluate the models in this table. Rather than focusing solely on word-level correctness, this metric evaluates how closely models approximate the target text by considering individual character-level differences. While dictionary-based and n-gram-based tools perform better on the filtered sets, LLMs, including both multilingual models, demonstrate performance that is relatively close to the original set, as shown in Table 5. According to the given results, the language-specific sequence-to-sequence model, T5-efficient-TR$_{small}$, achieves the best performance by improving the word correction score. It is also observed that the cross-lingual sequence-to-sequence model, FLAN-T5$_{small}$, exhibits lower performance on both the original and filtered datasets. Although cross-lingual encoder-decoder models demonstrate slightly reduced accuracy and word correction performance on the filtered set, owing to their broader capacity for handling diverse linguistic structures, mT5$_{small}$ still surpasses the encoder-only language-specific model in word correction. As the best-performing model, T5-efficient-TR$_{small}$ corrects sentences with the highest similarity, as measured by the Levenshtein distance. Moreover, mT5, as a powerful cross-lingual sequence-to-sequence model, along with encoder-only models, follows T5-efficient-TR$_{small}$ in terms of Levenshtein similarity.

### Comparison of models across different noise types

To assess the denoising capabilities of models with respect to various noise types, Table 7 presents the word-level correction performance for each noise type across LLMs. According to the table, T5-efficient-TR$_{small}$ outperforms all LLMs for each type of noise. Based on the table, it is evident that models are more capable of handling replacement noises than other types, while deletion noise proves to be the most challenging noise type to recover from. T5-efficient-TR$_{small}$ corrects only half of the typos, while it almost completely corrects all replacement errors. LLMs also appear to be more successful at handling insertion noises. As the lowest-performing sequence-to-sequence model,

FLAN-T5$_{small}$ exhibits relatively similar performance across most noise types, except for deletion noise. That is, this model demonstrates the poorest performance, primarily due to its inability to effectively correct deletion-type errors. Similarly, the encoder-only LLMs also appear to struggle with deletion noises, though they are capable of handling errors involving missing vowels.

## CONCLUSION

In this study, the spelling correction task is addressed for the Turkish language, which consists of many words that can be derived from the same root, as it is an agglutinative language. In addition to its potential applications in text-based platforms such as text editors, search engines, and email services, spelling mistake-free content is also crucial for existing LLMs in other NLP tasks as well. Therefore, the article focuses on the context-aware correction problem rather than traditional edit distance and dictionary-based approaches that consider similar words without regard to their surrounding context. By leveraging the performance of LLMs across a wide range of tasks, the study aims to explore LLMs for this downstream task on a novel dataset. With this goal, much of the work on understanding encoder-only and encoder-decoder models demonstrates how they perform in text correction, particularly when compared to traditional dictionary and n-gram based tools. A comprehensive comparison has been presented to provide insights into the performance of context-free and context-aware approaches, including LLMs, for spelling correction tasks. The evaluation utilizes the novel NoisyWikiTr dataset, comprising two distinct types of noise: uniformly generated noise and realistic noise to simulate user-behavioral errors. The results reveal that while dictionary-based tools perform better on uniformly noised texts, they struggle with realistic errors, highlighting their limitations in handling natural misspellings. Contextual models considerably outperform context-free approaches, with LLMs, particularly sequence-to-sequence models, demonstrating strong performance in handling realistic noise. Additionally, language-specific models, such as T5-efficient-TR$_{small}$, consistently outperform cross-lingual models, such as the powerful mT5$_{small}$, highlighting the advantage of specialized models for specific languages. In summary, these findings highlight the superiority of contextual models and the power of LLMs in the given specific task. As future work, autoregressive models, such as the GPTs, will be considered for this task, with a focus on addressing grammatical errors as well.

## ACKNOWLEDGEMENTS

OpenAI's GPT-4 was used for minor language refinements under the supervision of the author. It did not contribute to the research in terms of scientific content.

### Funding

This work was supported by TUBITAK ULAKBIM's High Performance and Grid Computing Center (TRUBA). Numerical calculations for training the models presented in

this article were partially performed using resources provided by TRUBA. The funders had no role in study design, data collection and analysis, decision to publish, or preparation of the manuscript.

### Grant Disclosures
The following grant information was disclosed by the authors:
TUBITAK ULAKBIM's High Performance and Grid Computing Center (TRUBA).

### Competing Interests
The authors declare that they have no competing interests.

### Author Contributions
- Ceren Guzel Turhan conceived and designed the experiments, performed the experiments, analyzed the data, performed the computation work, prepared figures and/or tables, authored or reviewed drafts of the article, and approved the final draft.

### Data Availability
The raw Turkish Wikipedia sentences are available at Hugging Face: https://huggingface.co/datasets/erenfazlioglu/turkishwikipedia2023.

For this study, these sentences were preprocessed and then corrupted by introducing noise to create the NoisyWikiTr dataset, which is available at GitHub and Zenodo:

- https://github.com/cgturhan/trspell.
- Guzel Turhan, C. (2025). NoisyWikiTr [Data set]. Zenodo. https://doi.org/10.5281/zenodo.15281473.

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
