# Peer review of "Leveraging large language models for spelling correction in Turkish"

_PeerJ Computer Science, doi:10.7717/peerj-cs.2889_

## Round 0.1 · original submission · Major Revisions

Both reviewers value the work in this paper, but also believe that substantial revisions are needed before it can be considered for publication, including: (i) need to flesh out technical detail to facilitate replication of the study, (ii) improve the justification for the use of synthetic data as well as discuss the limitations of doing this, (iii) there needs to be a more thorough discussion of the findings and their implications, which is somewhat lacking in depth at the moment, (iv) figures and tables need improving.

Please see reviewer comments for other concerns.

Reviewer 1 ·

Basic reporting

The paper explores the use of Large Language Models (LLMs) for spelling correction in Turkish, an agglutinative language posing unique challenges for such tasks. It introduces a novel dataset, NoisyWikiTr, containing synthetic spelling errors. The study compares BERT-based encoder-only models and T5-based encoder-decoder models against traditional tools and contextual models. It highlights the superiority of context-aware approaches and finds that encoder-only models outperform encoder-decoder models for this task.

Pros:
+ The introduction of NoisyWikiTr fills a significant gap by providing a large-scale benchmark tailored for Turkish, enhancing the reproducibility and future research in this domain.
+ The study benchmarks several traditional and advanced tools, offering valuable insights into the performance differences between context-free and context-aware models.
+ The work successfully demonstrates the advantages of leveraging contextual information for spelling correction in Turkish.

Cons:
- The synthetic nature of the NoisyWikiTr dataset, while comprehensive, may not fully represent the complexity of natural misspellings, limiting real-world applicability.
- The dataset is entirely synthetically generated, which may not fully reflect real-world misspelling patterns and could affect generalizability.
- While the paper evaluates several models, it omits comparisons with recent state-of-the-art multilingual LLMs beyond the ones included (e.g., newer versions of GPT or similar). Also, other related LLM-related works and correction models can be discussed, like LLM for complex language understanding, LLM for addressing multilingual tasks other than English, correction-tailored model.

Experimental design

The experimental design is well-structured, but there are areas where it could be improved:
1. Real-World Validation:
• The study relies exclusively on synthetic data, which may not capture the complexity of real-world errors. Incorporating evaluations on naturally occurring misspellings (e.g., from social media or user-generated content) would strengthen the findings and demonstrate broader applicability.
• Suggested Improvement: Include a subset of real-world noisy data or validate the model on an existing dataset like #Turki$hTweets.
2. Baselines and Comparisons:
• While traditional tools are included, the implementation of some baselines (e.g., Hunspell) might not be fully optimized for Turkish. Additionally, newer multilingual models are not included.
• Suggested Improvement: Optimize traditional baselines for Turkish and include comparisons with recent multilingual LLMs, such as newer versions of GPT or BLOOM.
3. Error Analysis:
• The paper lacks a detailed analysis of failure cases, which could provide insights into linguistic challenges or model limitations.
• Suggested Improvement: Conduct a qualitative error analysis, breaking down types of errors (e.g., morphological, contextual) to better understand model performance.
4. Hyperparameter Selection:
• The hyperparameter tuning process is not detailed, which raises questions about the robustness of the reported results.
• Suggested Improvement: Provide a more detailed explanation of hyperparameter optimization steps and justify the chosen settings.
5. Generalizability:
• The results are limited to Turkish, but the findings could benefit from demonstrating applicability to other agglutinative languages.
• Suggested Improvement: Extend experiments to a related language, such as Finnish or Hungarian, to validate the generalizability of the approach.

Summary of Improvements:

• Add real-world noisy datasets for validation.
• Optimize baseline implementations and include newer multilingual models.
• Provide an error analysis with insights into failure cases.
• Detail hyperparameter tuning strategies.
• Extend to other agglutinative languages to enhance generalizability.

Validity of the findings

The findings are generally valid and supported by the experimental results; however, some aspects could be improved to enhance the robustness of the conclusions:
1. Dataset Representativeness:
• The NoisyWikiTr dataset is synthetic, which raises concerns about its ability to capture real-world linguistic variability and misspelling patterns.
• Suggestion: Validate the findings using a real-world dataset to ensure that the reported performance translates to practical applications.
2. Overgeneralization:
• The conclusion that encoder-only models (e.g., BERTurk) outperform encoder-decoder models (e.g., T5) may not hold universally, as it is primarily based on synthetic data and a single language.
• Suggestion: Test the approach on other agglutinative languages or cross-linguistic datasets to confirm the generalizability of this conclusion.
3. Statistical Robustness:
• The paper does not provide statistical significance tests for performance differences between models.
• Suggestion: Include statistical significance testing to validate whether observed differences are meaningful and not due to random chance.
4. Confounding Factors:
• The study evaluates models trained on different datasets with potentially varying pretraining quality and scale, which could confound the results.
• Suggestion: Normalize the comparisons by training or fine-tuning models on the same dataset to isolate the impact of architecture.
5. Applicability to Downstream Tasks:
• While the paper focuses on spelling correction, the broader implications for NLP tasks reliant on error-free text are not sufficiently discussed.
• Suggestion: Demonstrate how improved spelling correction impacts downstream NLP tasks, such as sentiment analysis or machine translation.

Summary of Improvements:

• Validate findings on real-world data and other languages.
• Perform statistical significance tests for performance metrics.
• Ensure fair comparisons by controlling dataset and pretraining variables.
• Discuss broader implications for downstream NLP tasks.

Additional comments

NA

Cite this review as

Reviewer 2 ·

Basic reporting

- The English quality requires significant improvement. There are multiple instances of grammatical errors, typos, and awkward phrasing throughout the manuscript. The terminology is not consistent, making it harder to understand. (See my detailed review for specific line examples). A thorough revision by a professional English editor is crucial.

- The literature review is adequate for the NLP aspect but needs a stronger rationale for the research, especially in context of health informatics. It should establish the relationship with health informatics better and how those methods are useful for text mining in biomedical datasets. The selection of references appears somewhat arbitrary, and many lack direct DOI links and should be formatted according to the journal guidelines.

- The overall structure of the article is generally acceptable. However, the figures are not clear enough and lack detail. The tables need more comprehensive metrics, statistical information (such as standard deviations), and a much clearer description. Table 1 should include the distribution of noise, Table 3 should contain more hyperparameters. The raw data is shared, but the noising is not clearly justified.

- The paper generally presents relevant results, but the hypotheses are not clearly articulated at the beginning. Furthermore, the results section is very limited and lack in-depth analysis. The "results and discussion" should be combined into one section to properly explain each result.

- he paper lacks clear definitions of some important terms and metrics, especially for the deep learning architectures, and does not provide theorems, mathematical proofs, or much explanation in the method sections, which should be included in this type of research. The math formulas need to be explained in much more detail.

Experimental design

- The study aligns with the journal's focus on computer science and its application in NLP.
- The research question is defined, but the significance of the task in the bigger picture of health informatics is not emphasized enough. The specific rationale for focusing on Turkish is presented, but the unique challenges of the specific task should be articulated in more detail and justified. The need for robust spelling correction for downstream NLP tasks is mentioned, but the connection to health informatics is not explicit. How will this research be used in health informatics, for example, analysis of biomedical text? This is missing.
- The authors present data and experiments but lack rigor and technical clarity. There is not enough information for a reader to be able to replicate the methods exactly, especially regarding the hyperparameter tuning, training strategies, data preprocessing and cleaning techniques, and use of various libraries, such as Spacy. The noising strategy needs better justification with statistical parameters. The synthetic nature of the generated data should be thoroughly discussed and why the 10% noising was chosen. The models’ fine-tuning strategy is unclear. The experimental settings should include more details about the resources used for the research. The experimental setups should be justified. The details for the multi-head attention mechanisms and the Transformers architecture, and the implementation details are too limited
- The method descriptions lack enough detail. For example, the generation of the synthetic dataset is very brief and needs more information on the data cleaning, and sampling. The selection of different models is not very clear, why these specific libraries were chosen is not defined. The details of the underlying models, layers, activation functions, optimizers, etc., are missing. The “subword prediction models” are not well-described. The way masking is performed and what are the OOV words and how the masking is performed in Spacy needs more explanation. All of the figures need much more detail, specific layer description, and explanations.

Validity of the findings

- Meaningful replication encouraged where rationale & benefit to literature is clearly stated: While applying LLMs for spelling correction is not novel, the focus on Turkish and the creation of the dataset is a valuable contribution. The paper needs a much better justification for why this research is important. The rationale for this task in a real-world scenario needs to be explained. How these models will be used in real-world healthcare is not articulated and explained.
- The data is provided but synthetic. The dataset construction, especially the noising, needs much stronger justification and statistical analysis. The experiments need more ablation studies, confidence intervals, and error analyses. The metrics used in the paper are very limited. The comparison across different methods needs more explanation and justifications.
- The conclusions do not fully align with the presented results and are too limited. For example, the claim that encoder only models outperform encoder-decoder models needs more clarification on the reasons and why is this happening. The discussion on why the n-gram models perform better should be better explained. The study does not explore other LLM models. The discussion should not just focus on accuracy and word error rate, it should explore F1 score, precision, recall, and other sentence similarity scores. The conclusion that Jamspell shows the best performance should have more information on that model and why is it happening. The limitation of the studies should be articulated in much more detail, specifically, the generalization of the model should be mentioned. The paper should address why cross-lingual LLMs are performing poorly. It does not specify what parameters were optimized, what were the loss functions, and many other aspects that make it difficult to draw a strong conclusion

Additional comments

The overall structure is acceptable, but figures and tables need work. Figure captions need better description.

The paper would be significantly strengthened with additional experiments, ablation studies, and a much more in-depth discussion of the results.

Cite this review as

---

## Round 0.2 · accepted · Accept

I appreciate the authors for carefully revising the paper following reviewer suggestions. One of the original reviewers and myself have reviewed these changes, and come to the decision that it can be accepted in its current form.

While the reviewer suggested two references to be included, these should be considered optional.

Reviewer 1 ·

Basic reporting

The current version seems better, but the authors can discuss some recent related works, for example, using LLMs for complex understanding and correct tasks [1], and contextual spelling correction with LLMs [2].


[1] Can chatgpt understand too? a comparative study on chatgpt and fine-tuned bert. CoRR preprint 2023.
[2] Contextual spelling correction with large language models. In IEEE Automatic Speech Recognition and Understanding Workshop (ASRU) 2023.

Experimental design

NA

Validity of the findings

NA

Additional comments

NA

Cite this review as